# Development of Guar Gum-Pectin-Based Colon Targeted Solid Self-Nanoemulsifying Drug Delivery System of Xanthohumol

**DOI:** 10.3390/pharmaceutics14112384

**Published:** 2022-11-05

**Authors:** Mahesh Hanmantrao, Sourabh Chaterjee, Rajan Kumar, Sukriti Vishwas, Vancha Harish, Omji Porwal, Mohammed Alrouji, Othman Alomeir, Sharif Alhajlah, Monica Gulati, Gaurav Gupta, Kamal Dua, Sachin Kumar Singh

**Affiliations:** 1School of Pharmaceutical Sciences, Lovely Professional University, Phagwara 144411, India; 2Department of Pharmacognosy, Faculty of Pharmacy, Tishk International University, Erbil 4401, Iraq; 3Department of Medical Laboratories, College of Applied Medical Sciences, Shaqra University, Shaqra 11961, Saudi Arabia; 4Department of Pharmacy Practice, College of Pharmacy, Shaqra University, Shaqra 11961, Saudi Arabia; 5Faculty of Health, Australian Research Centre in Complementary and Integrative Medicine, University of Technology Sydney, Ultimo, NSW 2007, Australia; 6School of Pharmacy, Suresh Gyan Vihar University, Mahal Road, Jagatpura, Jaipur 302017, India; 7Department of Pharmacology, Saveetha Dental College, Saveetha Institute of Medical and Technical Sciences, Saveetha University, Chennai 602105, India; 8Uttaranchal Institute of Pharmaceutical Sciences, Uttaranchal University, Dehradun 248007, India; 9Discipline of Pharmacy, Graduate School of Health, University of Technology Sydney, Ultimo, NSW 2007, Australia

**Keywords:** xanthohumol, solid self-nanoemulsifying drug delivery system, guar gum, colon targeted delivery system, quality by design

## Abstract

Present study deciphers development of oral polysaccharide-based colon targeted solid self-nanoemulsifying drug delivery system (S-SNEDDS) of xanthohumol (XH). Several studies have shown that XH has anti-inflammatory and antioxidant properties, suggesting that it could be a good candidate for the treatment of colorectal diseases (CRD). Despite its potential, XH has a low aqueous solubility. As a result, its bioavailability is constrained by the dissolution rate. The liquid (L)-SNEDDS was constituted using Labrafac PG as oil, Tween 80 as surfactant and Transcutol P as co-surfactant. The L-SNEDDS was then adsorbed onto the surface of guar gum and pectin and developed into S-SNEDDS powder. Ternary phase diagram was used to optimize the process of developing L-SNEDDS. The formulation showed mean droplet size of 118.96 ± 5.94 nm and zeta potential of −19.08 ± 0.95 mV and drug loading of 94.20 ± 4.71%. Dissolution studies carried out in medium containing rat caecal contents (RCC) represented the targeted release of S-SNEDDS powder. It was observed that S-SNEDDS showed less than 10% release XH in initial 5 h and rapid release occurred between the 5th and 10th hour. Results of cytotoxicity studies revealed good cytotoxicity of XH loaded S-SNEDDS for Caco2 cells as compared to raw-XH.

## 1. Introduction

Xanthohumol (XH) is a prenylated flavonoid extracted from the female flowers of the hops plant (*Humulus lupulus* L.), which is primarily found in Germany and China [1]. XH is chemically known as 30-[3,3-dimethyl allyl]-20,40,4-trihydroxy-60-methoxychalcone [2]. The main constituent of the hop plant, which belongs to the Humulus genus and is a member of the Cannabaceae family, is XH. Because of its aroma and bitter taste, it is one of the ingredients used in beer [3]. This herbal drug shows promising anti-inflammatory [4], antioxidant [5], and anti-cancer [6] properties that can be used in the treatment of CRD.

The oral route is the most preferred route for the administration of drugs due to its advantages such as ease of administration, controllable dosage regimen, flexibility of both solid and liquid formulation, controllable and sustainable delivery, and patient compliance [7]. In order for an oral drug to have a therapeutic local effect in CRD, it must remain in its original form until it reaches the colon [8]. However, in the case of oral administration, the harsh acidic conditions of the upper gastrointestinal tract (GIT) and the presence of gastric enzymes cause denaturation of the dosage form, lowering its efficacy [9]; peristaltic movement, as well as shear stress caused by gastric enzymes in the GIT, causes mechanical degradation of the oral drug, resulting in poor site specificity [10]. As a result, for an oral dosage form to be effective, it must reach the colon without degradation, which can be accomplished by formulating an oral CTDS. Oral CTDS offers a significant advantage as compared to traditional dosage forms. The CTDS bypasses first-pass metabolism, allowing for localized and systemic drug delivery. In the upper GIT, all chemical and enzymatic degradation is avoided, and drug retention time is increased [11].

One of the most important factors in an oral formulation of a drug is its aqueous solubility, but XH, despite having the aforementioned beneficial effects, has poor aqueous solubility [12]. As a result, a dosage form is required in order to effectively solubilize the drug XH. In past years, various techniques were used to overcome the poor aqueous solubility of the drugs such as formation of complexes [13], nanoparticles [14,15,16], lipospheres [17,18,19,20], solid dispersions [21,22,23], solid state transformation [24], co-crystals [25,26,27], nanostructured lipid carriers [28] and vesicular delivery [29,30,31], etc. However, they have gained limited success and failed to become universal. Additionally, most of them lack commercialization due to failure during scale-up, instability or toxicity. In recent years, SNEDDS has been used successfully to overcome these challenges. SNEDDS is a novel drug delivery system in which an isotropic mixture of natural or synthetic oils, surfactants (ST), cosurfactants (CST) and drug molecules are used to form oil-in-water emulsion by agitation followed by dilution in aqueous media. SNEDDS offers a greater surface area for the dissolution and absorption of the drug. Both L-SNEDDS and S-SNEDDS enhance the penetrability and permeability of poorly aqueous soluble drugs, prevent the degradation of the drug in the gastrointestinal environment, and increase the oral bioavailability and dissolution [32]. These SNEDDS are solidified with various solid carriers such as Syloid XDP 3150 (SXDP) and aerosil 200 (A-200) [33]. To make the SNEDDS colon, targeting polysaccharides are used, such as guar gum (GUG) and pectin (PTN). These polysaccharides may lead to an increase in the colon targeting of SNEDDS formulation. As a result, a single delivery system can provide a dual benefit, which is improved XH solubility and formulation that can be carried unchanged to the colon due to the presence of polysaccharides, resulting in effective colon targeting. Polysaccharides are long chain polymeric carbohydrates or monosaccharides that are branched by glycoside linkage and are primarily derived from plants, animals, and microorganisms [34]. Polysaccharides such as pectin, guar gum, chitosan, fructan, polysaccharides derived from algae, and polysaccharides derived from other natural microorganisms have been shown to be safe and effective in the treatment of CRD [35]. Polysaccharides are mostly metabolized in the colon by colonic bacteria because they act as prebiotics (feed) for gut microbiota, which break down polysaccharides into simple saccharides and release the entrapped drug. Furthermore, polysaccharides can only be metabolized at alkaline pH and not at acidic pH. As a result, when colonic microbes use it, it leads to better breakdown in the large intestine. Polysaccharide also has significant advantages such as low cost, low toxicity, and high biocompatibility [36]. As a result, an effort has been made in this study to develop an oral colon-targeted polysaccharide-based SNEDDS loaded with XH for effective CRD treatment.

## 2. Materials and Methods

### 2.1. Materials and Equipments

XH was obtained as gift sample from Hopsteiner Trading (Zhuhai) Co., Ltd., Zhuhai, China. GUG, A-200, polyethylene glycol (PEG) 200, Tween 80 (T80), PTN, lactose monohydrate (LM), magnesium stearate (MgS), microcrystalline cellulose PH102 (MCC) was purchased from Lobacheime Pvt. Ltd., Mumbai, India. SXDP was gifted by Grace GmbH and Co KG, Mumbai, India. Peanut oil, cotton seed oil, eucalyptus oil, olive oil, castor oil was purchased from Global Merchants, Navi Mumbai, India. Hydrochloric acid (HCl) was acquired from Central Drug House, New Delhi, India. Labrafac PG (LPG), Labrafil M1944CS (LMCS) and Transcutol P (TP) were gifted by Gattefosse, Mumbai, India. Capmul MCM EP/NF (CMCM) was gifted by Abitec Corp., Mumbai, India. Methanol and ortho-phosphoric acid (OPA) High Performance Liquid Chromatography (HPLC) grade were acquired from Rankem, Mumbai, India. Distilled water (DW) was used throughout the study. The instruments used in this study were the same as reported in our previous study [37].

### 2.2. Methods

#### 2.2.1. Solubility Studies

A solubility study was conducted to determine which solubilizer would provide the best solubility for the drug. A known amount of XH (5 mg) was added to the separate test tubes containing individual oils (1 mL) (peanut oil, cotton seed oil, eucalyptus oil, olive oil, castor oil, LPG, LMCS, CMCM and Cremophor) and ST (PEG 200, PEG 400, T80 and TP), respectively. Into a mixture of 1 mL of oil, ST, and CST, a known excess amount of drug was added and vortexed for 5 min. The test tubes were then agitated in a shaking water bath for 48 h at 37 ± 2 °C. Following equilibration, in order to remove the undissolved XH from the saturated solution, all samples were centrifuged for 15 min at 10,000× *g*. Accurately measured supernatants were diluted with ethanol and n-hexane and the concentration of XH was determined using an HPLC system [38].

#### 2.2.2. Formulation of L-SNEDDS

Data of solubility studies revealed that LPG, T80 and TP were suitable for the development of L-SNEDDS formulation as these components have the highest solubility of XN. Here, LPG was used as oil, T80 and TP were used as ST and CST, respectively. L-SNEDDS were formed by varying the ratios of LPG, T80, and TP. LPG was used as O_mix_ and T80 and TP were used as S_mix_, with the ratios varying between 1:1, 1:2, and 2:1. The O_mix_ and S_mix_ ratios were varied from 1:9 to 9:1. Using these ratios, a total of 27 SNEDDS formulations were prepared. Depending upon the visual examination of the formulations, they were named as SNEDDS if they appeared as clear, transparent emulsion; SMEDDS, if they appeared as translucent emulsion. Out of 27 formulations, only 5 formulations, showed clear transparent emulsion. The data was entered into the software Triplot software version 4.1.2 (Todd Thompson Software version 4.1) in order to generate a ternary phase diagram which represents the area in which we can develop the clear L-SNEDDS formulations region. The composition of L-SNEDDS formulations shown in Table 1. For all batches, the SNEDDS pre-concentrate (1 mL) was first prepared by vortex mixing the oil, ST and CST in the specified ratio for 15 min. Following that, XH (10 mg) was added to each of the prepared pre-concentrates. After that, the mixture was vortexed for another 10 min with a vortex mixer. All the pre-concentrates loaded with XH were diluted with 250 mL distilled water at 750 rpm and the temperature was maintained 37 ± 0.2 °C for 10 min.

#### 2.2.3. Thermodynamic Stability Studies of L-SNEDDS Prototypes

To evaluate the thermodynamic stability, L-SNEDDS were placed at different temperature conditions. The L-SNEDDS were placed at 4 °C and 40 °C (Heating cooling cycles) and at −21 °C and +25 °C (freeze thaw cycles) for 48 h to assess the thermal stability. In order the check the centrifugation stability the diluted L-SNEDDS (1:100 in distilled water) were centrifuged for 15 min at 3500× *g*, then observed visually for the phase separation [39]. The prepared L-SNEDDS were diluted (1:250 in distilled water) and then gradually heated on a temperature-controlled water bath in order to check the cloud point. The initial temperature of water was 25 °C and was then raised at a rate of 5 °C/min. The point/temperature where the cloudiness appears in the diluted L-SNEDDS was considered as the cloud point and the temperature was noted [40].

#### 2.2.4. Formulation of Solid (S) SNEDDS

##### Determination of Drug Loading Factor (L_f_)

The drug L_f_ value was used to determine the liquid absorption capacity of the solid carriers. A higher L_f_ value indicates that less carrier is required to provide free-flowing powder [41]. The L_f_ value of some of the commonly used carriers for solidifying L-SNEDDS, such as LM, PTN, A-200, SXDP, MgS, MCC and GUG was calculated. L-SNEDDS were placed in a mortar, and weighed amounts of solid carriers were added and gently blended by pestle to calculate $L_{f}$. The addition of solid carriers was continued until the powder was free flowing. The formula to calculate $L_{f}$ is given in Equation (1),
(1)$$L_{f}=\frac{W_{1}}{W_{c}}$$
where, W_1_ = Weight (Wt) of the liquid; W_c_ = Wt of the carrier required to solidify.

##### Angle of Repose (θ) (AOR)

AOR (θ) was determined to check flow properties by using the standard funnel method. In summary, the funnel was fixed 4 cm above the surface of the bench. S-SNEDDS powder blends that have been prepared were poured one at a time through this funnel [21]. The radius (r) and height (h) of the pile were measured from the base. The formula to calculate the angle of repose is given in Equation (2),
(2)$$\theta ={\;\tan}^{-1}\left(\frac{h}{r}\right)$$

##### Adsorption to Solid Carriers and Colon Targeting Formulation of S-SNEDDS

Adsorption into solid carriers can produce free-flowing powder systems from L-SNEDDS formulations [33]. The adsorption technique has the advantage of achieving high content uniformity. Polysaccharides such as PTN and GUG have been shown to be both safe and effective in the treatment of IBD. Because polysaccharides act as prebiotics (food) for gut microbiota, which break down polysaccharides into simple saccharides and release the entrapped drug, they are mostly metabolised in the colon by colonic bacteria [42]. Hence, to make the SNEDDS colon targeting it was adsorbed on polysaccharides and solid carriers. S-SNEDDS were prepared by mixing the L-SNEDDS containing XH with PTN, GUG, SXDP. In a mortar, L-SNEDDS were applied gradually over the carrier. To ensure uniform distribution of the formulation, the mixture was vigorously blended and homogenised after each addition. The resulting formulation was dried at room temperature. 

#### 2.2.5. Characterization of SNEEDS

##### Pre-Compression Properties of S-SNEEDS Powders

Bulk density (BD), flow rate, Carr’s compressibility index (CCI) and tapped density (TPD) were all measured as pre-compression parameters. All of the investigations were conducted in the manner as per described by Beg et al., 2016 [43]. 

##### Bulk Density (ρb)

S-SNEDDS powder was accurately weighed and poured through a graduated measuring cylinder and bulk density was calculated by using Equation (3): (3)$$\rho b=\frac{M}{\mathrm{Vb}}$$

Here, M = Wt. of the powder, Vb = bulk volume.

##### Tapped Density (ρt)

S-SNEDDS powder was weighted accurately and taken in a measuring cylinder. The cylinder was tapped 100 times. Using Equation (4), tap density was determined.
(4)$$\rho t=\frac{M}{\mathrm{Vt}}$$

Here, M = Wt. of the blend; Vt = volume occupied by the blend in the cylinder.

##### Carr’s Compressibility Index (CCI)

Carr’s compressibility index (CI) was calculated using the formula given in Equation (5):(5)$$\mathrm{CCI}=\frac{\mathrm{pt}-\mathrm{pb}}{\mathrm{pt}}\times 100$$

##### Drug Loading (DL)

Optimized batches of L-SNEDDS and S-SNEDDS were prepared and tested for % DL. The XH-loaded SNEDDS (10 mg) were diluted in distilled water (500 mL) while being stirred at 500 rpm [37]. The 5 mL of sample was centrifuged for 15 min at 10,000× *g* to remove the undissolved XH. The clear supernatant was collected, filtered through a 0.2 m syringe filter, and the concentration was calculated using HPLC at 368 nm. The % DL was determined using the formula in Equation (6):(6)$$\mathrm{DL}\left(\%\right)=\frac{\mathrm{Area}\;\mathrm{of}\;\mathrm{the}\;\mathrm{test}\;\mathrm{drug}\;\mathrm{present}\;\mathrm{in}\;\mathrm{SNEDDS}}{\mathrm{Area}\;\mathrm{of}\;\mathrm{the}\;\mathrm{known}\;\mathrm{standard}}\times 100$$

##### Droplet Size (DS) and Zeta Potential (ZP) Analysis

The Malvern zeta sizer nano ZS90 (Malvern Instruments Ltd., Worcestershire, UK) was used to determine the DS, PDI, and ZPO of L-SNEDDS and S-SNEDDS. At 25 °C, recordings were made in disposable polystyrene cells with a 50 mV laser at a fixed angle of 90°. A 0.1 mL sample of SNEDDS was diluted with 100 mL of distilled water. The dilution (1 mL) was taken in the sample cell and analysed [44]. 

##### Robustness to Dilution and pH Change

Despite the changing pH and volume of the GI fluids from stomach to colon, it is important that SNEDDS maintain their size as they move through the GIT. To test this, S-SNEDDS were diluted in four different media: water, phosphate buffer (0.2 M, pH 6.8), 0.1 M acetate buffer (pH 4.5), and 0.1 N HCl (pH 1.2). In each of the four media, S- SNEDDS (1 mL) were diluted to 10, 100, 250, and 900 times. All of the diluted samples had the same mean DS and phase separation. Each study was done in triplicate, and the mean value was recorded [45].

##### Differential Scanning Calorimetry (DSC) and Powder X-ray Diffraction (PXRD) Studies

Thermal analysis of raw XH, LPG, TP, T80, PTN, GUG, SXDP and S-SNEDDS powder was completedby using a DSC Q200 TA. Samples were processed as per the method described by Inugala et al. [38]. In brief, first samples were sealed individually in aluminium pan and then heated from 0 to 300 °C. The temperature was raised at a rate of 10 °C/min [38]. An X-ray diffractometer was used to record the PXRD patterns of raw XH, PTN, GUG, SXDP and S-SNEDDS powder. As a radiation source, a copper line was used. The sample was hit by X-ray beams with a voltage of 40 kV and a current of 40 mA. Samples were scanned at a constant rate of min^−1^ throughout a 2 range of 345 degrees [46].

##### Microscopic Analysis

Transmission Electron Microscopy (TEM)

TEM was used to detect the droplet morphology of the optimised L-SNEDDS and S-SNEDDS formulations. The optimised SNEDDS formulation (100 µL) was diluted with double distilled water (10 mL). On a carbon-coated copper grid, a drop of emulsion was placed in order to create a thin film for negative staining, and then with the help of filter paper extra solution was removed. After 10 min, one drop of phosphotungstic acid (2% *w/v*) solution was dripped for about 1 min on the copper grid, and the excess solution was removed. The grid was allowed to dry naturally, and the sample was analysed using TEM [47].

Scanning Electron Microscopy (SEM)

Surface morphological analysis of raw XH and XH in S-SNEDDS was carried out by SEM. Raw XH, PTN, GUG, SXDP and S-SNEDDS powder were all analyzed using SEM. The samples were secured with conductive tape to a metallic stub (12 mm diameter) prior to the analysis. Supra 35 V P was the data station used (Oberkochen, Zeiss, Germany). The voltage in the range of 5 to 25 kV was used in order to accelerate the electrons [48].

##### In Vitro Dissolution Studies Using Rat Caecal Contents

As the delivery of the drug was expected into the colon region, it is necessary to mimic the colon condition at the time of dissolution study. However, the conventional dissolution media cannot provide such conditions. To overcome this challenge, the rat caecal contents were added into the conventional dissolution medium for mimicking human and rodent colonic microflora. The caecal contents were taken from the healthy rats of ongoing protocol approved from Institutional Animal Ethics Committee (LPU/IAEC/2021/86). For initial 2 h, dissolution of XH loaded L-SNEDDS, S-SNEDDS powder, raw XH was performed in a 0.1 N HCL medium having pH 1.2 using a basket apparatus (USP Type-I). The formulation (1 g) was filled in size 000 capsule shells and subjected for dissolution. The temperature was maintained at 37 ± 0.2 °C and the rotation of baskets was fixed at 100 ± 4 rpm. Thereafter, pH 1.2 was modified to pH 6.8 with 0.2 M phosphate buffer and the study was continued up to 24 h for L-SNEDDS, raw-XH and S-SNEDDS. 

In addition, in order to check the effect of microbiota on the release behaviour of S-SNEDDS an additional study was carried out. After completion of 5 h as per the above-mentioned procedure, rat caecal contents were added, and the 7.2 pH maintained by adding 1–2 mL of NaOH. Nitrogen gas was passed to maintain the anaerobic conditions. Prior to adding the rat caecal contents it was diluted with phosphate-buffered saline (pH 6.8) to acquire desired concentrations. XH equivalent to 10 mg was present in the SNEDDS formulations subjected to dissolution studies. These SNEDDS were placed in hard gelatin capsule shells of size “000”. At predetermined time intervals, samples were withdrawn up to 24 h and a fresh medium in equal volume to the withdrawn sample volume was added in order to maintain the sink conditions. The withdrawn samples were centrifuged for 10 min at 10,000× *g*, supernatant was collected and filtered with the help of syringe filters (0.2 µm). The filtered samples were further analyzed at 368 nm on HPLC in order to check the release of the drug [49]. 

##### Release Kinetic Study

Raw XH, S-SNEDDS in RCC, S-SNEDDS without RCC and L-SNEDDS were subjected to an in vitro release study by taking an accurate quantity of samples containing XH equivalent to 20 mg. The release study was performed using dissolution apparatus (USP type II) with the following conditions: phosphate buffer pH 6.8 as dissolution media maintained at 37 ± 0.5 °C temperature, at a stirring speed of 50 rpm. All the samples were weighed accurately and filled into a hard gelatin capsule of size ‘0’ and kept in the stainless-steel capsule sinker and then subjected to dissolution. The samples (5 mL) were withdrawn at different time intervals (0, 1, 2, 3, 4, 5, 6, 7, 8, 10, 12, 18, and 24 h) and a fresh sample equivalent to the withdrawn sample is added. The withdrawn sample was passed through the membrane filter (0.22 µm). The filtered samples were centrifuged for 15 min and the supernatant was collected and injected to the HPLC for determining the concentration of XH. The release mechanism of XH from SNEDDS were studied by applying various kinetic models such as zero order, first order, Higuchi, Hixson–Crowell and Korsmeyer–Peppa’s models.

##### Cytotoxicity Study Using MTT Assay

The 3-[4,5-dimethylthiazole-2-yl]-2,5-diphenyltetrazolium bromide (MTT) assay was carried out on human colorectal adenocarcinoma (Caco2) cell lines. The Caco2 cell lines (ATCC HTB-3) culture in PPMi-1640 medium (sigma) were used for this study. The study was carried out as per the procedure reported by Corrie et al. (2022) with slight modifications [49]. The cells were supplemented with 10% and 20% fetal bovine serum (FBS) and 1% penicillin-streptomycin solution (antibiotic) for nutrition. The cell was stored at a temperature of 37± 0.03 °C with incubation of CO2 (5%). The cells were seeded in a 96-well plate with a density of 15,000 cells in each cell well [50]. At the end of 24 h of incubation, the medium was replaced with different test solutions that included blank solution (PBS 200 µL, S1), Placebo SNEDDS (200 µL, S2), XH suspension (200 µM, S3), and XH loaded S-SNEDDS (200 µM, S4). These cells suspended in a new medium with different test solutions were incubated for 6 h. This was followed by a replacement of the old medium with a fresh one after 24 h. In brief, MTT (5 mg) was dissolved in 1 mL of PBS and added to each well containing test, blank and standard solution. The cells were incubated for 3 h (37 ± 0.03 °C, 5% CO_2_). After incubation, the old medium was replaced with dimethyl sulphoxide (100 µL per well). The plates were shaken for 10 min and optical density of the samples were measured at 570 nm using a microplate reader (Bio-Rad labs, Hercules, CA, USA). The medium was aspirated at the end of 12 h and 24 h post treatment. The percentage of survival of cells was estimated using the procedure in Equation (7) [47].
(7)% Cell viability=[mean optical density of test group ÷mean optical density of control group]×100

##### Stability Studies

An accelerated stability study was conducted for selected S-SNEDDS. S-SNEDDS were placed in the stability chamber at 40° ± 2 °C/75% ± 5% R.H. for three months. After completion of 3 months, samples were analyzed for DS, DL (%), drug precipitation, and AOR and data was compared with the same parameters of the fresh samples. In order to evaluate the DS, DL and the drug precipitation, the S-SNEDDS were reconstituted in distilled water and the liquid formulation was analyzed as per their aforementioned procedures.

##### Statistical Analysis

The experimental data were expressed as mean ± standard deviation (SD). Statistical analysis of the obtained data was completed by applying analysis of variance or Tukey’s multiple comparison test. The difference was considered to be significant if the value of *p* found to be <0.05. For performing this whole statistical analysis, GraphPad Prism version 7.0 (GraphPad Software Inc., San Diego, CA, USA) was used.

## 3. Results and Discussion

### 3.1. Solubility Studies

The results of solubility study of XH in oils, ST and CST are displayed in Table 1. In case of oils tested, XH had the highest solubility in LPG (480.39 ± 38.43 µg/mL). Among CST, raw XH had the highest solubility in TP (525.31 ± 36.7 µg/mL). To develop a clear nanoemulsion, the oil, ST, CST and oil to ST/CST ratio must be carefully chosen. To achieve this, surfactant should have a hydrophilic-lipophilic balance (HLB) value greater than 10 in order to form oil in water emulsion. In case of ST, XH had the highest solubility in T80 (81.18 ± 6.49 µg/mL). The HLB value of T80 is also high, i.e., 15, so it was selected as a surfactant. 

### 3.2. Construction of Ternary Phase Diagram

A total of 27 SNEDDS (Table 2) prototypes were prepared and observed visually. A titration of homogeneous liquid mixtures of oil, ST and CST with water was used to develop the ternary phase diagrams. S_mix_ was prepared in various volume ratios, such as 1:1, 1:2, and 2:1. Oil and specific S_mix_ ratios were mixed in different volume ratios ranging from 1:9 to 9:1 in order to develop SNEDDS formulation. It was observed that transparency of the formulation gets decreased, but DS get increased as the oil ratio increased. The reason behind this can be that an increase in oil phase concentration leads to increased lipid concentration in the formulation, and the concentration of ST was not sufficient to maintain homogeneous dispersion of nano size droplets due to increased interfacial tension at the oil–water interface [50]. On the other hand, when the oil concentration is low, i.e., below 30%, and the concentration of ST is high, i.e., above 40%, leads to a decrease in interfacial tension at the oil–water interface. This results in the formation of homogeneous dispersion of nano size droplets with a clear and transparent appearance. Out of 27 formulations, only five formulations (F1, F2, F10, F19 and F20) were found to be clear and termed as SNEDDS preparations (Figure 1). This indicated that the concentration of LPG was important in reducing the droplet size of SNEDDS. Similarly, among the ST (T80) and CST (TP), the higher the concentration of TP played a significant role in reducing DS to the nano level. The prepared formulations (F1, F2, F10, F19 and F20) were placed for 48 h at room temperature, and after completion of the 48 h time period, these formulations were observed visually for clarity and stability. Even after 48 h, all of the tested emulsions remained clear and transparent. As a result, these formulations have been chosen for further research.

### 3.3. Identification of Droplet Size, Thermodynamic Stability and Drug Loading of Selected L-SNEDDS

All the selected formulations (F1, F2, F10, F19 and F20) were found to be stable thermodynamically. No phase separation or precipitation was seen throughout alternate temperature cycles (4 °C and 40 °C), freeze–thaw cycles (−21 °C and +25 °C), or centrifugation at 10,000× *g*, showing that the formulations were stable. The results are shown in Table 2. The DS ranged from 108.1± 5.41 nm (F1) to 134.5 ± 6.72 nm (F20). % DL of XH in L-SNEDDS ranged from 46.42 ± 2.3% (F1) to 94.20 ± 4.7% (F10). The cloud point of all five SNEDDS formulations was between 77 to 95 °C, indicating that they were stable enough to travel through the GIT after being diluted with gastric fluids [47].

The findings revealed that the presence of more T80 and TP played a significant role in increasing and decreasing DL and DS in emulsions [38]. As shown in Table 2, the results of formulation F10 (1 mL) containing a 1:9 ratios of O_mix_ [LPG (100 µL, i.e., 10%) to S_mix_ (T80:TP ratio was 1:2, i.e., T80 was 30% and TP was 60%) showed the smaller DS, the highest DL, and the best stability against thermodynamic stress and centrifugation. As a result, it was chosen for further investigation.

### 3.4. Formulation and Characterization of S-SNEDDS

The formulation “F10” was solidified using various carriers. The values of $L_{f}$ for solidifying L-SNEDDS (0.1 g) prepared using MCC, GUG, PTN, SXDP, MgS, LM, A-200, and GUG:PTN:SXDP (1:1:1) were 0.38 (0.26 g), 0.13 (0.75 g), 0.15 (0.65 g), 1.11 (0.09 g), 0.20 (0.49 g), 0.12 (0.81 g), 0.62 (0.16 g) and 0.41 (0.24), respectively. A higher $L_{f}$ value indicates that less carrier was used to convert liquid to free-flowing powder. When compared to other carriers, SXDP had the highest $L_{f}\;$value, followed by a combination of polysaccharides (i.e., GUG-PTN). It was discovered that when GUG or PTN were used alone, their $L_{f}\;$value was lower than when they were combined. LM has the lowest$\;L_{f}$ value. The AOR for PTN, A-200, SXDP, MgS, MCC, LM, GUG and GUG:PTN:SXDP (1:1:1) were 40.5, 20.76, 19.59, 30.3, 53.0, 18.4, 29.8, and 24.62 degrees, respectively. The results showed that the L_f_ and AOR for SXDP and GUG:PTN:SXDP (1:1:1) were almost identical, indicating that the combination of GUG:PTN could be used as a carrier.

SNEDDS were solidified by using adsorption techniques, in which L-SNEDDS formulation was added onto inert solid carrier, i.e., SXDP. To develop the colon targeted SNEDDS, polysaccharides such as GUG:PTN:SXDP (1:1:1) were added and mixed it in a blender and dried. DS, ZP, pre-compression characteristics and % DL studies were performed on these S-SNEDDS powder. All of the S-SNEDDS powder passed the thermodynamic and centrifugation stability tests as no phase separation and no drug precipitation observed. Table 3 displays the pre-compression properties, % DL, ZP and mean DS of all prepared S-SNEDDS. 

The pre-compression properties of the carriers used to develop S-SNEDDS were highly variable. These include flow rate, AOR, bulk density, tapped density and Carr’s compressibility index. With respect to the carriers used, the mean DS and PDI also varied. The flow rate was found to be between 0.87 g/s (LM) and 3.16 g/s (SXDP). The AOR ranged from 19.50^o^θ (SXDP) to 53.0^o^θ (MCC). BD ranged from 0.13 g/cm^3^ (A-200) to 0.44 g/cm^3^ (GUG), whereas TPD ranged from 0.22 g/cm^3^ [GUG:PTN:SXDP (1:1:1)] to 0.60 g/cm^3^ (GUG). The CCI ranged from 55.55 (MCC) to 18.18 (SXDP). The pre-compression properties of various carriers differ due to differences in their physicochemical properties and L_f_. SXDP showed very good flow rate, BD, TP, and TPD when compared to other solid carriers used as it is fluffy and has a high surface area. Following SXDP, S-SNEDDS prepared with GUG:PTN:SXDP (1:1:1) as a carrier manifest good flow rate and density values. This meant that S-SNEDDS made with GUG:PTN:SXDP (1:1:1) as carriers had nearly identical flow behaviour to S-SNEDDS made with SXDP. The main advantage of using GUG:PTN:SXDP (1:1:1) as a solid carrier was its CCI value. The CCI value of GUG:PTN:SXDP (1:1:1) was found to be lower than the CCI value of SXDP 3150. A lower CCI value indicated that the carrier’s compaction properties were excellent. During solidification when liquid droplets and solid carriers were dispersed into the aqueous medium it leads to an increase in the DS. Following dilution, the carrier with the highest solubility in the aqueous medium produced a clear transparent emulsion, followed by the carrier with the lowest solubility in the medium. The size of the droplets increased further after adsorption onto solid carriers due to increased adherence of liquid droplets with their solid carriers [33]. The DL in all formulations was found to be greater than 85%. As a result, based on the overall performance of S-SNEDDS prepared with GUG:PTN:SXDP (1:1:1) as carrier, i.e., high flow rate, good AOR, low CCI value, good DL, high ZP, and smallest DS, it was deemed the best carrier to solidify emulsions.

### 3.5. PXRD Analysis

Raw XH, GUG, PTN, SXDP, and (GUG:PTN:SXDP) S-SNEDDS powder PXRD patterns were recorded, and the results are shown in Figure 2a. Similar to DSC data, the crystalline nature of XH and amorphous nature of SXDP were observed. The amorphous halo pattern was obtained for GUG, which indicated an absence of any definite crystalline peak. PTN exhibited some sharp peaks followed by an absence of sharp peaks, indicating a semi-crystalline or disordered crystalline structure. PXRD revealed the amorphous nature of (GUG:PTN:SXDP) S-SNEDDS formulations, implying that the crystalline drug was effectively dissolved in the L-SNEDDS formulation. Because of the apparent uniform blending of porous carriers, polysaccharide, and L-SNEDDS preconcentrate, it is important to highlight that uniform mixing of porous carriers and L-SNEDDS preconcentrate improved the amorphous nature of formulations, allowing complete adsorption of dissolved XH on carriers’ surface. To gain a better understanding, the DSC thermograms were correlated with the PXRD data.

### 3.6. DSC Analysis

XH exhibited a sharp crystalline endo-thermic peak at 158 °C (Figure 2b), whereas the excipients used in the formulations (LPG, T80, TP, SXDP and GUG) exhibited an amorphous nature as evidenced by their halo pattern, with no sharp peaks in PXRD. PTN exhibits a slight sharp peak at 182 °C, indicating its melting point. The (GUG:PTN:SXDP) S-SNEDDS formulation’s amorphous halo profile revealed full drug absorption in the liquid formulation as well as liquid adhesion to the porous carriers and polysaccharide. The change in XH from crystalline to amorphous S-SNEDDS formulation or molecularly dispersed state is advantageous for improving its solubility and dissolution profiles. Furthermore, XH is adequately stabilized by the oil surfactant–co-surfactant framework. 

### 3.7. Microscopic Analysis

#### 3.7.1. SEM Analysis

Figure 3a–e shows SEM images of raw XH, GUG, PTN, SXDP and (GUG:PTN:SXDP) S-SNEDDS powder. XH revealed crystalline particles with an irregular shape and numerous particles clumped together (Figure 3a). GUG had a flake-like cylindrical structure (Figure 3b). PTN showed a rough and globular surface structure, indicating that it is semi-crystalline in nature (Figure 3c). The surface of SXDP was discovered to be highly porous and rough (Figure 3d). The S-SNEDDS powder appeared as smooth surfaced particles with porous and spherical in shape indicating the presence of adsorbed liquid SNEDDS preconcentrate on the porous surface of carriers, particularly GUG and SXDP (Figure 3e). The obtained results were found to be similar to those obtained from PXRD and DSC studies for the developed (GUG:PTN:SXDP) S-SNEDDS formulation. The microscopic analysis findings were consistent with previous findings and matched the S-SNEDDS formulation results from PXRD and DSC studies.

#### 3.7.2. TEM Analysis

TEM images of XH-loaded L-SNEDDS and S-SNEDDS are showed in Figure 4a,b. Surface morphology and DS diameter were examined in these images. There was no evidence of droplet aggregation, and the droplets were smooth and circular in shape. The mean DS of XH-loaded L-SNEDDS powder was 112.9 ± 5.64 nm, and the DS of XH-loaded S-SNEDDS powder was 118.7 ± 5.94 nm. The TEM of S-SNEDDS also revealed a coating over the spherical droplet, suggesting that the polysaccharides GUG and PTN were uniformly coated over the L-SNEDDS. Such coating was not observed around LSNEDDS droplets.

### 3.8. Effect of Change in Dilution Volume and pH on Droplet Size

Dilution alters the profile of drug release in vivo, and the possibility of drug precipitation reduces drug solubility. The effect of change in pH and dilution on DS of the optimized formulation is summarized in Figure 5. Results were represented that there was no significant change in DS after changing the pH and volume of the formulation. The dilution volume XH loaded (GUG:PTN:SXDP) S-SNEDDS were found to be robust for oral delivery of the formulation due to a non-significant change in DS with pH and dilution. Furthermore, consistent DS after dilution procedures suggests that in vivo drug release will be uniform.

### 3.9. In Vitro Dissolution Study

The immediate release pattern was observed for the XH loaded in L-SNEDDS as more than 75% of the drug was released in the first hour. The drug release was 100% dissolved in the first 3 h from L-SNEDDS; whereas only 10% release was observed for raw XH in the first 5 h and 23.87% at the end of 24 h, which is indicative of a poor dissolution rate of XH owing to its lipophilicity. This indicated about 4.19-fold enhancement in dissolution of XH upon loading into SNEDDS as that of raw XH. The results are shown in Figure 6. When the release of polysaccharide-based S-SNEDDS of XH was analyzed in the presence of rat caecal contents, less than 10% of the drug was released in the first 5 h (Figure 6), which suggests that drug release in the upper GIT is restricted due to the presence of polysaccharide matrix, which could have slowed drug release in the absence of microbiota. However, when the same S-SNEDDS were entered into an anaerobic culture medium containing live faecal microbiota, a sudden jump in drug release from 5th hour (8.23%) to 12th hour (86.88%) was observed, indicating rapid polysaccharide digestion by the microbes present in the RCC. The increase in drug release after 6 h in the presence of microbiota indicates that the drug is released after polysaccharide degradation, which is only possible in the lower GIT and colon. Many studies have been published previously on the dissolution testing of polysaccharide-based oral colon targeted delivery system, in which RCC were used to conduct dissolution testing and were found to be successful in demonstrating the role of polysaccharides in limiting drug release for the first 5 h (i.e., upper GIT).

Interestingly, when the same S-SNEDDS were subjected to medium without rat caecal contents after 5 h, only a 32.21% release was observed in 24 h. This showed that the developed polysaccharide-based formulation is a microbial-triggered colon-targeted delivery system.

### 3.10. Evaluation of Release Kinetics

The in vitro drug release mechanisms were studied by applying various kinetic models such as zero order, first order, Higuchi, Hixson–Crowell and Korsmeyer–Peppa’s models. The parameters of the kinetic models were represented in Table 4. The various values of correlation coefficient (R^2^) indicated the best fit to the model. S-SNEDDS in RCC, S-SNEDDS without RCC and L-SNEDDS were best fitted with the Korsmeyer–Peppa’s model with R^2^ nearer to 1, and n value < 0.45 showing the release pattern was following Fickian diffusion. The formulations shown the sustained release profile mainly due to the presence of polysaccharides. The drug release from the raw XH was followed by zero order kinetics and Korsmeyer–Peppa’s model. By observing the release profiles and mechanisms, the XH released from the S-SNEDDS in RCC showed good release of the drug with 98.11% in 24 h; whereas, XH from S-SNEDDS without RCC showed only 32.21%, which indicated the RCC has a major impact on enhancing the release of drug from the formulation. XH release from L-SNEDDS was faster, i.e., 100% drug was released within 4 h.

### 3.11. Cytotoxicity Studies

To investigate the anticancer potential of XH-loaded S-SNEDDS, the caco-2 cell line study was carried out [51]. XH at a dose of 200 µM showed good cytotoxicity to caco-2 cells. Hence, this concentration was used for cell line studies. The results were noted at 12 h and 24 h post treatment of cells and found that all the test samples (S3 and S4) showed a time-dependent toxicity to caco-2 cells. However, the cell viability of above 90% indicated no cytotoxicity with samples S1 and S2, indicating that the excipients used in this study were safe. It is important to highlight here that XH (S3) showed cellular toxicity of 62.56 ± 4.87% and 48.33 ± 3.88% at the end of 12 h and 24 h. In a similar way, XH-loaded S-SNEDDS (S4) also showed good cytotoxicity to Caco2 cells as it can be evident from the fact that only 43.90 ± 3.98% and 31.98 ± 3.10% cells were found viable at the end of 12 and 24 h. This showed a significant (*p* < 0.05) enhancement in cytotoxicity of XH upon loading it into S-SNEDDS. This increase in cytotoxicity of XH-loaded S-SNEDDS (S4) was about 1.42-fold and 1.51-fold at the end of 12 h and 24 h, respectively, as that of cytotoxicity of raw-XH (S3). The results of cell viability are given in Figure 7. 

### 3.12. Stability Studies

The mean DS of aged (GUG:PTN:SXDP) S-SNEDDS powder was 122.65 ± 6.13 nm and 118.96 ± 5.94 nm for fresh (GUG:PTN:SXDP) S-SNEDDS powder, respectively. Similarly, the DL for fresh S-SNEDDS was 96.84 ± 4.81%, whereas it was 93.11 ± 4.65% for aged S-SNEDDS. The AOR for fresh S-SNEDDS was 22.62 ± 1.23, and the AOR for aged S-SNEDDS was 24.27 ± 1.21. The non-significant difference in the above results indicated that the prepared formulations were stable [52,53,54]. 

## 4. Conclusions

A XH SNEDDS was successfully developed using LPG, T80, and TP as formulation components. To optimize the formulation variables, Triplot software (version 4.1.2 by Todd-Thompson) was used. The optimized batch of L-SNEDDS was then solidified using an adsorption technique with hydrophobic solid carriers and polysaccharides to form S-SNEDDS. This was followed by a thorough investigation involving pre-compression properties, in vitro dissolution and stability studies. The flow and compression properties were found to be carrier dependent. When under different stress conditions such as TDS and FTC, the formulated S-SNEDDS, prepared by using SXDP as a hydro-phobic carrier and polysaccharides such as GUG and PTN, provided nanoemulsions with unchanged DS and drug release. In vitro studies revealed that the S-SNEDDS were significantly superior to raw XH. SEM, DSC, and PXRD analysis confirmed that the crystalline form of XH was converted to the amorphous form in S-SNEDDS formulation containing SXDP as a carrier and GG, PTN as a polysaccharide. The successful formulation of S-SNEDDS powder revealed that GG and PTN have excellent compaction properties. The lack of any significant difference between old and fresh formulation indicated that GG and PTN had the potential to provide stable formulations. XH-loaded S-SNEDDS showed 1.42-fold and 1.51-fold higher cytotoxicity at the end of 12 h and 24 h, respectively, in comparison with the cytotoxicity of raw-XH. This indicated the potential of SNEDDS in treating colon cancer as that of its raw form. The current investigation also provides the opportunity for future production-scale exploration of this combination. The positive outcomes of this study indicated good potential of XH in treating CRD, such as colon cancer, upon delivering them at colonic site, as well as opening new avenues for scientists for the formulation and development of nanoparticles of plant bioactives for treating diseases.

## Figures and Tables

**Figure 1 pharmaceutics-14-02384-f001:**
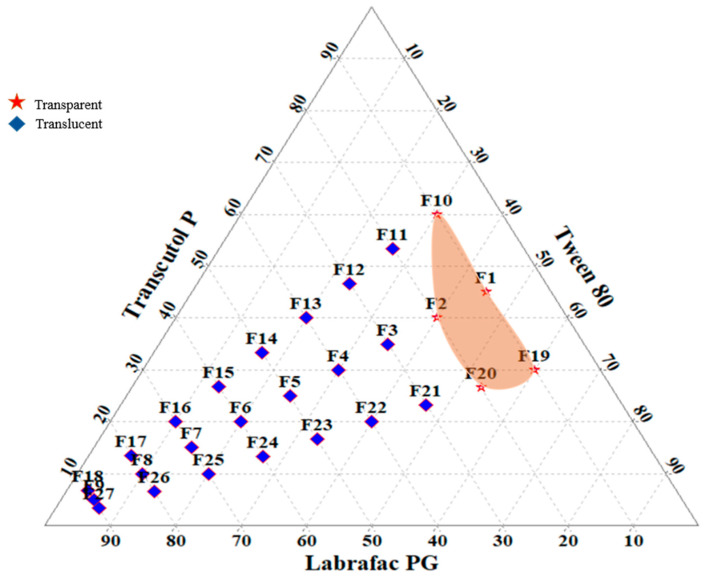
Ternary phase diagram of various L-SNEEDS prototypes.

**Figure 2 pharmaceutics-14-02384-f002:**
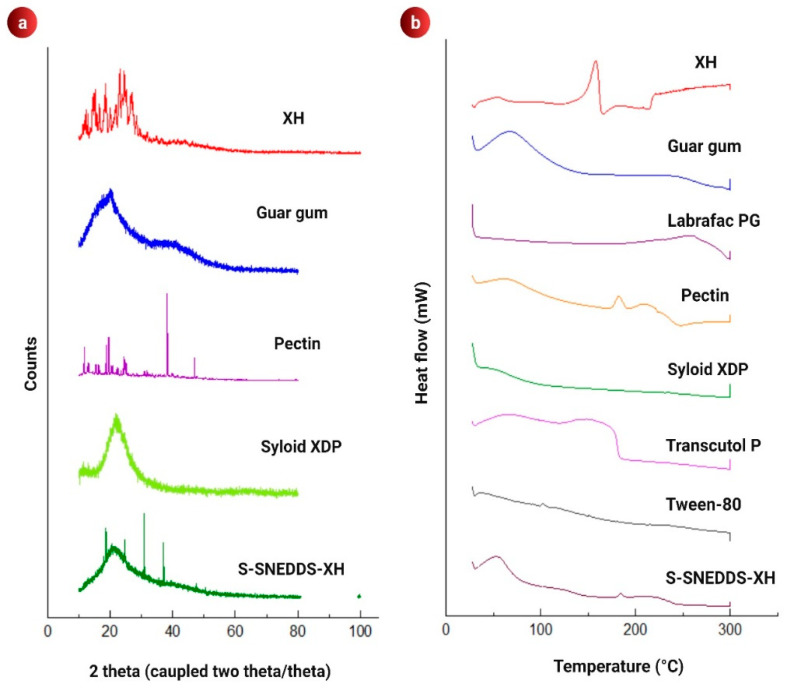
Results showing (**a**) overlay plot of PXRD patterns of XH, excipients and S-SNEDDS formulation; (**b**) DSC thermograms of XH, excipients and S-SNEDDS formulation.

**Figure 3 pharmaceutics-14-02384-f003:**
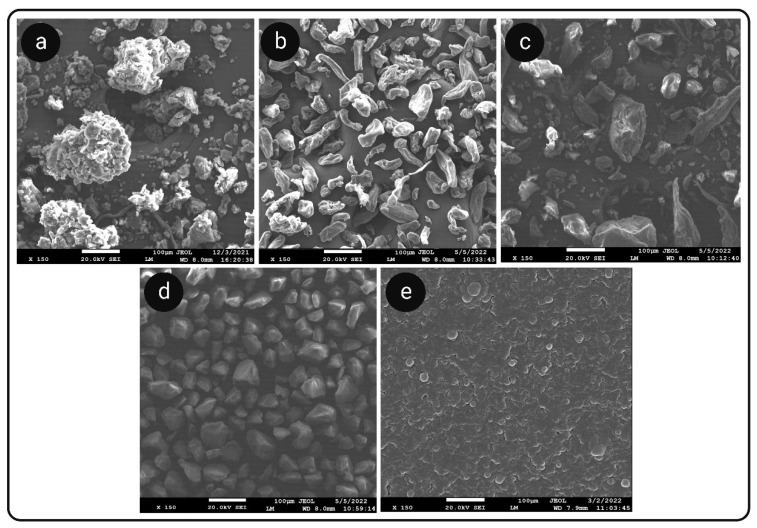
SEM images of (**a**) raw XH; (**b**) Guar gum; (**c**) Pectin; (**d**) Syloid XDP3150; (**e**) S-SNEDDS powder.

**Figure 4 pharmaceutics-14-02384-f004:**
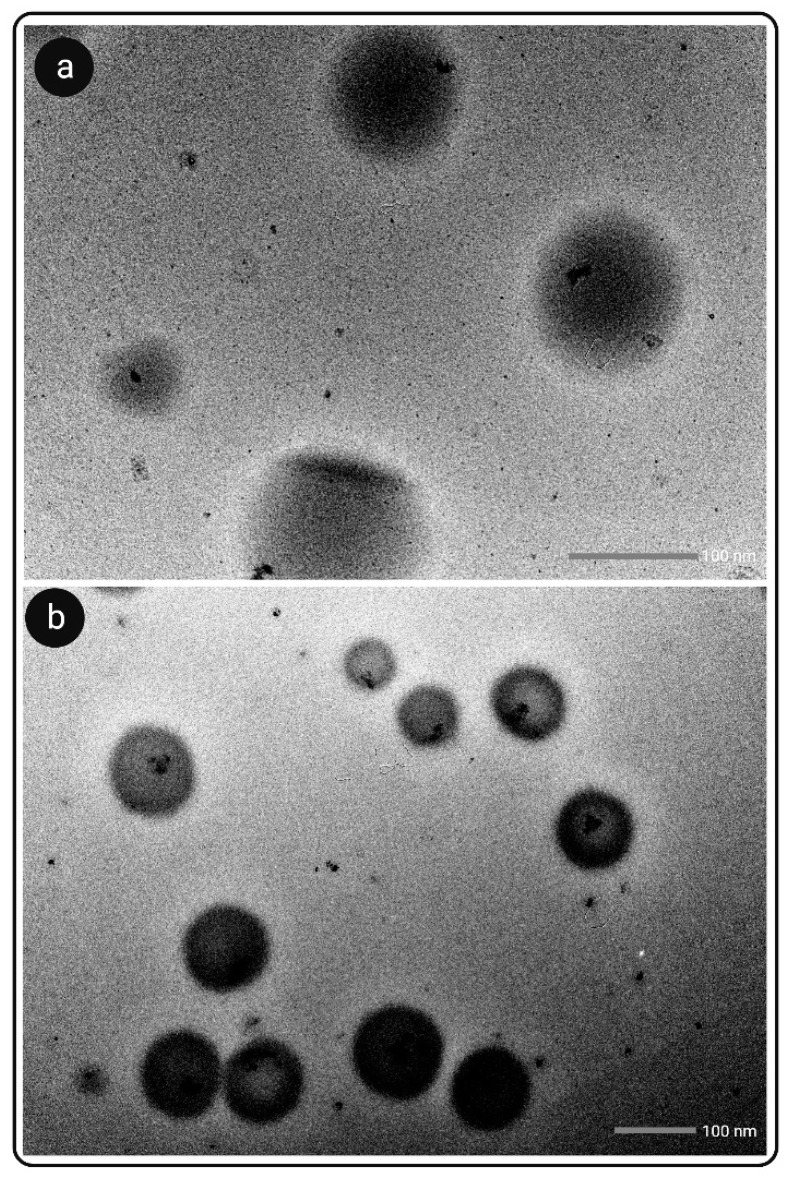
TEM images of (**a**) L-SNEDDS and (**b**) S-SNEDDS.

**Figure 5 pharmaceutics-14-02384-f005:**
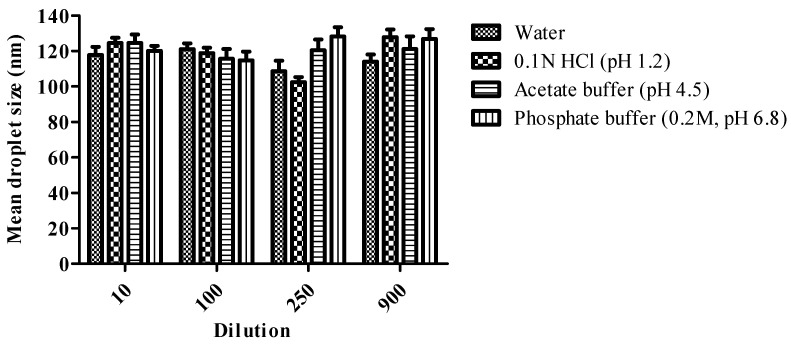
Effect of change in dilution volume and pH on mean droplet size (Mean ± s.d.; n = 3).

**Figure 6 pharmaceutics-14-02384-f006:**
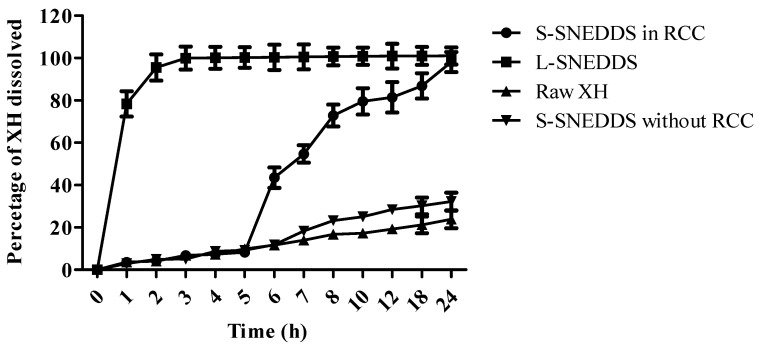
Results of dissolution studies (n = 6) for raw XH, L-SNEDDS, and S-SNEDDS in RCC and without RCC. RCC–Rat caecal contents (n = 6).

**Figure 7 pharmaceutics-14-02384-f007:**
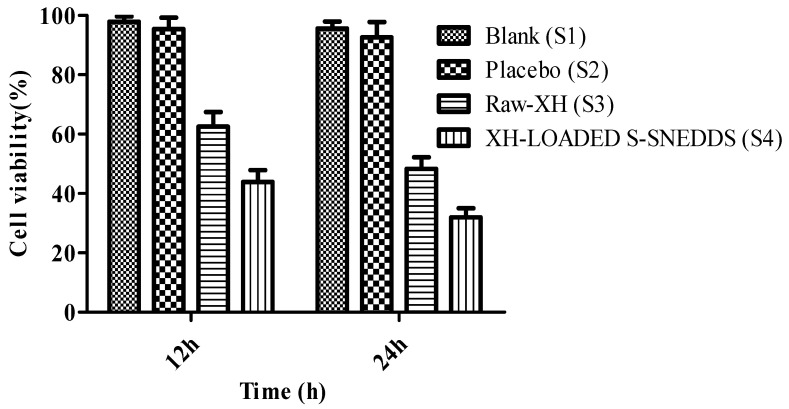
Results of cell viability (%) for blank, placebo, raw-XH and XH-loaded S-SNEDDS [number of replicates (n) = 3].

**Table 1 pharmaceutics-14-02384-t001:** XH solubility in different vehicles (data represented as mean ± S.D.; n = 3).

Vehicle	Solubility of Raw XH (µg/mL)
Olive oil	173.54 ± 10.46
Cottonseed oil	61.59 ± 3.07
Peanut oil	67.83 ± 4.75
Eucalyptus oil	200.02 ± 18.01
Castor oil	176.16 ± 15.85
Labrafil M1944 CS	82.56 ± 6.70
Capmul MCM	16.96 ± 1.17
Labrafac PG	480.39 ± 38.43
Cremaphor EL	113.30 ± 10.20
Transcutol P	525.31 ± 36.7
Tween 80	81.18 ± 6.49
PEG 200	228.65 ± 20.5

**Table 2 pharmaceutics-14-02384-t002:** Evaluation parameters (n = 5) and composition of selected batches of XH loaded L-SNEDDS (%*w/w*).

Formulation Code	Composition for L-SNEDDS (1 mL) (LPG:T80:TP) (%)	Droplet Size (nm) (Mean ± s.d.)	Polydispersity Index (PDI) (Mean ± s.d.)	Drug Loading (%) (Mean ± s.d.)	Cloud Point (°C) (Mean ± s.d.)	Appearance
F1	10:45:45	108.1 ± 5.41	0.29 ± 0.014	46.42 ± 2.32	77.61 ± 3.84	TP
F2	20:40:40	124.7 ± 6.20	0.36 ± 0.018	83.31 ± 4.16	89.23 ± 4.41	TL
F10	10:30:60	112.9 ± 5.64	0.23 ± 0.011	94.20 ± 4.71	95.88 ± 4.73	TP
F19	10:60:30	118.8 ± 5.94	0.41 ± 0.020	91.01 ± 4.55	94.61 ± 4.80	TP
F20	20:53:27	134.5 ± 6.72	0.33 ± 0.016	81.21 ± 4.06	91.57 ± 4.06	TL

TP—Transparent; TL—Translucent.

**Table 3 pharmaceutics-14-02384-t003:** Pre-compression characteristics of S-SNEDDS (n = 3, mean ± S.D.).

Components	Flow Rate (g/s)	Angle of Repose (θ)	Bulk Density (g/cm^3^)	Tap Density (g/cm^3^)	CCI	Drug Loading (%)	Mean Droplet Size (nm)	PDI	Zeta Potential (mv)
L-SNEDDS (F10)	NA	NA	NA	NA	NA	94.20 ± 4.71	112.9 ± 5.64	0.23 ± 0.01	−19.08 ± 0.95
		After adsorpion onto solid carriers							
S-SNEDDS (F10)									

MCC	2.74 ± 0.13	53.00 ± 2.65	0.20 ± 0.01	0.45 ± 0.02	55.55 ± 2.77	89.47 ± 4.47	143.21 ± 7.16	0.48 ± 0.02	−19.43 ± 0.97
GUG	1.23 ± 0.06	28.83 ± 1.44	0.44 ± 0.02	0.60 ± 0.03	26.60 ± 1.33	92.14 ± 4.60	157.95 ± 7.89	0.29 ± 0.01	−21.98 ± 1.09
PTN	2.01 ± 0.10	40.51 ± 2.02	0.43 ± 0.02	0.54 ± 0.02	20.37 ± 1.01	94.78 ± 4.73	134.46 ± 6.72	0.26 ± 0.01	−19.87 ± 0.99
SXDP	3.16 ± 0.15	19.50 ± 0.97	0.18 ± 0.009	0.24 ± 0.01	18.18 ± 0.90	95.26 ± 4.76	124.76 ± 6.23	0.22 ± 0.01	−18.56 ± 0.92
MgS	1.11 ± 0.05	30.31 ± 1.51	0.30 ± 0.01	0.44 ± 0.02	31.81 ± 1.59	87.43 ± 4.37	148.53 ± 7.42	0.36 ± 0.01	−21.87 ± 1.09
LM	0.87 ± 0.04	29.42 ± 1.47	0.42 ± 0.02	0.57 ± 0.02	26.31 ± 1.31	85.41 ± 4.27	159.51 ± 7.97	0.45 ± 0.02	−23.76 ± 1.18
A-200	2.89 ± 0.14	20.76 ± 1.03	0.13 ± 0.006	0.29 ± 0.01	55.17 ± 2.75	92.97 ± 4.64	139.24 ± 6.96	0.39 ± 0.01	−21.48 ± 1.05
GUG:PTN:SXDP (1:1:1)	3.03 ± 0.15	22.62 ± 1.23	0.17 ± 0.008	0.22 ± 0.01	22.74 ± 1.13	96.84 ± 4.81	118.96 ± 5.94	0.19 ± 0.009	−22.78 ± 1.13

**Table 4 pharmaceutics-14-02384-t004:** Results of release kinetic profiles of Raw XH, L-SNEDDS, S-SNEDDS without RCC and S-SNEDDS in RCC.

Model	S-SNEDDS in RCC					L-SNEDDS				
	R^2^	K	RMSE	AIC	BIC	R^2^	K	RMSE	AIC	BIC
Zero order	0.6108	3.43	1.6952	6.5922	6.6081	0.1646	1.62	2.4401	1.204	1.2153
First order	0.3625	7.36	2.5069	7.2182	7.2341	0.1129	3.7	1.3423	2.246	2.2573
Higuchi	0.4995	1.74	1.9223	6.7933	6.8092	4.5899	1.23	6.312	1.4511	1.4624
Krosmeyer Peppas	0.5233	4.64	2.4185	7.1607	7.1766	0.9857	1.33	1.2026	1.6187	1.63
Model	Raw XH					S-SNEDDS without RCC				
	R^2^	K	RMSE	AIC	BIC	R^2^	K	RMSE	AIC	BIC
Zero order	0.8576	9.9	2.6917	6.3089	6.4219	0.8357	1.49	4.4091	7.592	7.705
First order	0.1591	4.06	3.2298	1.8756	1.8869	0.1732	4.27	5.2706	2.0029	2.0142
Higuchi	0.7273	1.66	9.3761	9.5536	9.6666	0.5941	1.91	1.3733	1.0545	1.0658
Krosmeyer Peppas	0.9978	1.76	6.9537	8.7765	8.8895	0.9988	2.03	6.8756	8.7472	8.8602

## Data Availability

Not applicable.

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
