# Peer review of "Development of Guar Gum-Pectin-Based Colon Targeted Solid Self-Nanoemulsifying Drug Delivery System of Xanthohumol"

_pharmaceutics, 2022, doi:10.3390/pharmaceutics14112384_

Round 1

Reviewer 1 Report

Dear Authors,

You present very basic research with no scientific news without solving the therapeutic problem. In my opinion, the results of the study are non-application.

I think that supplementing the results with the effect on intestinal cells is necessary.

In my opinion, the novelty of the manuscript is low and so it may be published after major revision.

The list of the most important comments is as follows:

1.      The abbreviation S-SNEDDS and L-SNEDDS are too complicated for me, please simplify them.

2.      Please provide an abbreviation CRD in the abstract (line 46).

3.      More keywords increase the chances of being cited. I recommend increasing the number of keywords.

4.      Line 59 – 85 contain well-known knowledge. Etiopathogenesis and classic treatments are not needed in a journal with a high IF.

5.      The determination of the mechanism of release and the fitting to such mathematical models as zero-order, first-order, Higuchi, and Korsmeyer-Peppas are necessary.

6.      Please complete the bar in the SEM photos (Fig. 4).

7.      Please eliminate the typo in the description of fig. 5.

8.      Elimination of capital letters in references is necessary.

Best regards,

Review

Author Response

Dear Editor,

First of all, we would like to thank you to provide us the opportunity to revise this article. We would also like to express our sincere thanks to the learned reviewers for their sagacious suggestions. Their suggestions have really helped in improving the quality of manuscript. We went through the comments and added the required information. The corrections are done in track changes mode.

If still any further action is required, then please let us know. We will be happy to address that also.

Reviewer 1#

Comments 1: The abbreviation S-SNEDDS and L-SNEDDS are too complicated for me, please simplify them.

Response: The S-SNEDDS resemble solid-self nanoemulsifying drug delivery system and L-SNEDDS resemble liquid self-nano emulsifying drug delivery system. It is a general term used for SNEDDS and frequently used by researchers. Many of such studies are cited in the reference sections. Hence, I request the worthy reviewer to kindly consider this.

Comments 1: Please provide an abbreviation CRD in the abstract (line 46).

Response: Provided.

  1. More keywords increase the chances of being cited. I recommend increasing the number of keywords.

Response: As suggested, the number of keywords have been increased.

  1. Line 59 – 85 contain well-known knowledge. Etiopathogenesis and classic treatments are not needed in a journal with a high IF.

Response: Thanks for the valuable suggestion. We have deleted these sections.

  1. The determination of the mechanism of release and the fitting to such mathematical models as zero-order, first-order, Higuchi, and Korsmeyer-Peppas are necessary.

Response: As per the suggestions of the learned reviewer, we have applied the release kinetic mathematical models. The methodology is described is section 2.2.5.11 and results are described in section 3.10 and table 4.

  1. Please complete the bar in the SEM photos (Fig. 4).

Response: The error bars are provided in all the SEM and TEM images.

  1. Please eliminate the typo in the description of fig. 5.

Response: Corrected.

  1. Elimination of capital letters in references is necessary.

Response: Corrected.

Reviewer 2 Report

General comments

The manuscript entitled “Formulation and evaluation of guar gum-pectin based oral colon targeted solid self-nanoemulsifying drug delivery system of xanthohumol: A novel study” is an interesting work that proposes an oral delivery system for targeting xanthohomol, an antioxidant and anti-inflammatory agent, to the colon. The authors performed several in vitro tests to evaluate the technological properties of the optimized delivery system based on a self-nanoemulsifying formulation that support the usefulness of the investigated strategy.

However, the manuscript contains many inaccuracies that should be addressed.

Specific comments

Line 46. The meaning of the abbreviation CRD should be explained. The meaning of the abbreviations should be explained the first time they are cited in the text. Please, check throughout the manuscript and correct.

Line 54. A burst release occurs when a fast drug release is observed at the very beginning of the release process while the author state that a burst release occurred after 5 h. Therefore, the term “burst” is incorrect and should be replaced.

Line 70. The meaning of the sentence “Despite the fact that no single factor has been identified as the primary cause of the aforementioned diseases.” Is unclear. Please, rephrase.

Line 309. What amount of formulation did the authors use to fill size 0 capsules?

Line 329. The authors performed stability studies on solid SNEDDS by evaluating DS (droplet size), DL (drug loading) drug precipitation and AOR. It is unclear how could the authors determine DS and drug precipitation of a solid formulation that did not contain droplets or a liquid phase from which the drug could precipitate. Please, explain.

Line 362- 366- 372. F1, F2, F19, F19 and F20 should read F1, F2, F10, F19 and F20. Please correct.

In Table 2, the appearance of formulation F20 is missing. Please, add this information in Table 2.

Line 412. What are “micromeritics properties”? Please, explain.

Line 455. What is the meaning of “halo pattern”? Please, explain.

Line 461. The sentence “Figure 4b depicts the results” should be deleted, as there is no Fig. 4b in the manuscript and Fig. 2 b, which illustrates the results, has been already cited in the text.

The legend of figure 2 does not match the plots illustrated in the figure. Please, correct.

Line 484. The authors claim that in Fig. 3f the droplets were circular and uniform in diameter. Looking at Fig. 3f it is evident that the droplets have different diameters. Please, discuss the results according to the depicted data.

English should be carefully revised throughout the manuscript. There are several typos and grammar mistakes.

Author Response

Dear Editor,

First of all, we would like to thank you to provide us the opportunity to revise this article. We would also like to express our sincere thanks to the learned reviewers for their sagacious suggestions. Their suggestions have really helped in improving the quality of manuscript. We went through the comments and added the required information. The corrections are done in track changes mode.

If still any further action is required, then please let us know. We will be happy to address that also.

  1. Line 46. The meaning of the abbreviation CRD should be explained. The meaning of the abbreviations should be explained the first time they are cited in the text. Please, check throughout the manuscript and correct.

Response: Provided.

  1. Line 54. A burst release occurs when a fast drug release is observed at the very beginning of the release process while the author state that a burst release occurred after 5 h. Therefore, the term “burst” is incorrect and should be replaced.

Response: Corrected.

  1. Line 70. The meaning of the sentence “Despite the fact that no single factor has been identified as the primary cause of the aforementioned diseases.” Is unclear. Please, rephrase.

Response: Corrected.

  1. Line 309. What amount of formulation did the authors use to fill size 0 capsules?

Response: We used size “000” capsules. Size “0” capsule was written by mistake (section 2.2.5.10). 1g powder was added.

  1. Line 329. The authors performed stability studies on solid SNEDDS by evaluating DS (droplet size), DL (drug loading) drug precipitation and AOR. It is unclear how could the authors determine DS and drug precipitation of a solid formulation that did not contain droplets or a liquid phase from which the drug could precipitate. Please, explain.

Response: In order to evaluate the DS, DL and the drug precipitation, the S-SNEDDS were reconstituted in distilled water and the liquid formulation was analyzed. It is mentioned in section “2.2.5.12. Stability Studies”

  1. Line 362- 366- 372. F1, F2, F19, F19 and F20 should read F1, F2, F10, F19 and F20. Please correct.
  2. In Table 2, the appearance of formulation F20 is missing. Please, add this information in Table 2.

Response: Authors are thankful for such valuable suggestions. We have added the details in Table 2.

  1. Line 412. What are “micromeritics properties”? Please, explain.

Response: The word “micromeritics” is replaced by “pre-compression” throughout the manuscript.

  1. Line 455. What is the meaning of “halo pattern”? Please, explain.

Response: Halo pattern in PXRD resembles absence of any peak in the powder X-ray diffractogram, resembling amorphous nature of compound.

  1. Line 461. The sentence “Figure 4b depicts the results” should be deleted, as there is no Fig. 4b in the manuscript and Fig. 2 b, which illustrates the results, has been already cited in the text.

The legend of figure 2 does not match the plots illustrated in the figure. Please, correct.

Response: Corrected.

  1. Line 484. The authors claim that in Fig. 3f the droplets were circular and uniform in diameter. Looking at Fig. 3f it is evident that the droplets have different diameters. Please, discuss the results according to the depicted data.

Response: Corrected.

  1. English should be carefully revised throughout the manuscript. There are several typos and grammar mistakes.

Response: We have taken the help of an English expert and corrected the language throughout the manuscript.

Reviewer 3 Report

Hanmantrao M et al, show the develop of different oral drug delivery system based on polysaccaridhes for specific release xanthohumol. The manuscript should be rewritten as it is very long-winded in sections and lacking information in others.

It’s recommended a Major Revision of this manuscript.

The following revisions are requested:

-The title is long and ambiguous. Please remodulate it.

-The introduction is confusing, please rearrange to make it clearer.

-Figure 3, it’s difficult to read the scale bars of the SEM images, and in Figure 3f,g, scale bars missing. Please, improve it.

- Please, rearranging the results and discussion section seems too long-winded in the first paragraphs and slim in the last ones.

Author Response

Dear Editor,

First of all, we would like to thank you to provide us the opportunity to revise this article. We would also like to express our sincere thanks to the learned reviewers for their sagacious suggestions. Their suggestions have really helped in improving the quality of manuscript. We went through the comments and added the required information. The corrections are done in track changes mode.

If still any further action is required, then please let us know. We will be happy to address that also.

  1. The title is long and ambiguous. Please remodulate it.

Response: We have shortened the title as per the suggestions of learned reviewer.

  1. The introduction is confusing, please rearrange to make it clearer.

Response: We have deleted the general sections from the introduction sections related to etiopathogenesis of colorectal diseases.

  1. Figure 3, it’s difficult to read the scale bars of the SEM images, and in Figure 3f,g, scale bars missing. Please, improve it.

Response: Corrected as suggested.

  1. Please, rearranging the results and discussion section seems too long-winded in the first paragraphs and slim in the last ones.

Response: Corrected as suggested.

Round 2

Reviewer 1 Report

Dear Authors,

My most important point was to supplement the results with the effect on intestinal cells is necessary.

The manuscript should not have been published in a journal with such a high IF, the quality of the results and conclusions is not adequate for the journal.

Best regards,

Review 

Author Response

Comment 1: My most important point was to supplement the results with the effect on intestinal cells is necessary. The manuscript should not have been published in a journal with such a high IF, the quality of the results and conclusions is not adequate for the journal.

Response: Thanks for the valuable suggestion. We apologize that somehow we missed this comment to be addressed during first revision. As per the sagacious suggestions of learned reviewer, we have added the results of MTT assay that has been carried out on human colorectal carcinoma (Caco2) cells. This cell line is best fit for colorectal diseases. The methodology is described in section 2.2.5.12. and results are discussed in section 3.11 and Figure.7.

Reviewer 2 Report

The authors revised the manuscript properly.

Author Response

Authors are thankful to the learned reviewer to accept the revision.

Reviewer 3 Report

The paper can be accepted in the present form.

Author Response

(The authors gave the same response as above.)
